# Safety evaluation of the single-dose Ad26. COV2.S vaccine among healthcare workers in the Sisonke study in South Africa: A phase 3b implementation trial

Simbarashe Takuva[1,2☯*], Azwidhwi Takalani[1,3☯], Ishen Seocharan[4], Nonhlanhla Yende-Zuma[5], Tarylee Reddy[4], Imke Engelbrecht[6], Mark Faesen[6], Kentse Khuto[1], Carmen Whyte[1,2], Veronique Bailey[1], Valentina Trivella[6], Jonathan Peter[7], Jessica Opie[8], Vernon Louw[9], Pradeep Rowji[10], Barry Jacobson[11], Pamela Groenewald[12], Rob E. Dorrington[13], Ria Laubscher[4], Debbie Bradshaw[12], Harry Moultrie[14], Lara Fairall[15,16], Ian Sanne[6], Linda Gail-Bekker[17], Glenda Gray[18], Ameena Goga[19,20‡], Nigel Garrett[5,21‡], Sisonke study team[¶]

1 Vaccine and Infectious Disease Division, Fred Hutchinson Cancer Research Center, Seattle, Washington, United States of America, 2 School of Health Systems and Public Health, Faculty of Health Sciences, University of Pretoria, Pretoria, South Africa, 3 Department of Family Medicine and Primary Care, Faculty of Health Sciences, University of the Witwatersrand, Johannesburg, South Africa, 4 South African Medical Research Council, Durban, South Africa, 5 Centre for the AIDS Programme of Research in South Africa, Durban, South Africa, 6 Right to Care, Johannesburg, South Africa, 7 Division of Allergy and Clinical Immunology, Faculty of Health Sciences, University of Cape Town, Cape Town, South Africa, 8 Division of Haematology, Department of Pathology, Faculty of Health Sciences, University of Cape Town and National Health Laboratory Service, Groote Schuur Hospital, Cape Town, South Africa, 9 Division of Clinical Haematology, Department of Medicine, Faculty of Health Sciences, University of Cape Town and Groote Schuur Hospital, Cape Town, South Africa, 10 Milpark Hospital, Johannesburg, South Africa, 11 Department of Molecular Medicine and Haematology, Charlotte Maxeke Johannesburg Academic Hospital National Health Laboratory System Complex and University of the Witwatersrand, Johannesburg, South Africa, 12 Burden of Disease Research Unit, South African Medical Research Council, Cape Town, South Africa, 13 Centre for Actuarial Research, Faculty of Commerce, University of Cape Town, Cape Town, South Africa, 14 National Institute for Communicable Diseases, National Health Laboratory Service, Sandringham, South Africa, 15 Knowledge Translation Unit, University of Cape Town Lung Institute, Department of Medicine, University of Cape Town, Cape Town, South Africa, 16 King's Global Health Institute, King's College London, London, United Kingdom, 17 Desmond Tutu HIV Centre, University of Cape Town, Cape Town, South Africa, 18 South African Medical Research Council, Cape Town, South Africa, 19 Department of Paediatrics and Child Health, Faculty of Health Sciences, University of Pretoria, Pretoria, South Africa, 20 HIV Prevention Research Unit, South African Medical Research Council, Cape Town, South Africa, 21 School of Nursing and Public Health, Discipline of Public Health Medicine, University of KwaZulu-Natal, Durban, South Africa

☯ These authors contributed equally to this work.
‡ These authors are joint senior authors on this work.
¶ Membership of the Sisonke study team is provided in the Acknowledgements.
* simbataks1@gmail.com

**Data Availability Statement:** The data are owned by a third party (The National Department of Health, South Africa). The data underlying the

## Abstract

### Background

Real-world evaluation of the safety profile of vaccines after licensure is crucial to accurately characterise safety beyond clinical trials, support continued use, and thereby improve public confidence. The Sisonke study aimed to assess the safety and effectiveness of the Janssen Ad26.COV2.S vaccine among healthcare workers (HCWs) in South Africa. Here, we present the safety data.

results presented in the study may be made available upon request or application and the necessary ethical approvals. Contact information for data requests: Office of the Director-General, E-mail: DG@health.gov.za.

**Funding:** Funding for the Sisonke Study was provided by: The National Department of Health through baseline funding to the South African Medical Research Council (no grant number; websites: https://www.health.gov.za/ and https://www.samrc.ac.za/); the Solidarity Response Fund NPC (no grant number; website: https://solidarityfund.co.za/support/); The Michael & Susan Dell Foundation (no grant number; website: https://www.dell.org/); the ELMA Vaccines and Immunization Foundation (Grant number 21-V0001; website: https://www.elmaphilanthropies.org/vaccines); and the Bill & Melinda Gates Foundation (grant number INV-030342; website: https://www.gatesfoundation.org/). L-GB: declares honoraria for advisory roles from MSD, ViiV Health Care, and Gilead. JP: declares support by a career development award (grant no. K43TW011178-04) and financial support from the National Institutes of Health (award no. K43TW011178-02); the European Developing Clinical Trials Partnership (EDCTP2 Program supported by the European Union grant no. TMA2017SF-1981); and the SA Medical Research Council and National Research Foundation. The funders had no role in study design, data collection and analysis, decision to publish, or preparation of the manuscript.

**Competing interests:** I have read the journal's policy and the authors of this manuscript have the following competing interests: JP received speakers fees from Johnson and Johnson, and spouse is employed by Johnson and Johnson. The other authors have declared that no competing interests exist.

**Abbreviations:** AE, adverse event; CRF, case report form; EVDS, Electronic Vaccination Data System; FDA, Food and Drug Administration; GBS, Guillain-Barré syndrome; HCW, healthcare worker; SAE, serious adverse event; TTS, thrombosis with thrombocytopenia syndrome.

## Methods and findings

In this open-label phase 3b implementation study among all eligible HCWs in South Africa registered in the national Electronic Vaccination Data System (EVDS), we monitored adverse events (AEs) at vaccination sites through self-reporting triggered by text messages after vaccination, healthcare provider reports, and active case finding. The frequency and incidence rate of non-serious and serious AEs were evaluated from the day of first vaccination (17 February 2021) until 28 days after the final vaccination in the study (15 June 2021). COVID-19 breakthrough infections, hospitalisations, and deaths were ascertained via linkage of the electronic vaccination register with existing national databases. Among 477,234 participants, 10,279 AEs were reported, of which 138 (1.3%) were serious AEs (SAEs) or AEs of special interest. Women reported more AEs than men (2.3% versus 1.6%). AE reports decreased with increasing age (3.2% for age 18–30 years, 2.1% for age 31–45 years, 1.8% for age 46–55 years, and 1.5% for age > 55 years). Participants with previous COVID-19 infection reported slightly more AEs (2.6% versus 2.1%). The most common reactogenicity events were headache ($n = 4,923$) and body aches ($n = 4,483$), followed by injection site pain ($n = 2,767$) and fever ($n = 2,731$), and most occurred within 48 hours of vaccination. Two cases of thrombosis with thrombocytopenia syndrome and 4 cases of Guillain-Barré Syndrome were reported post-vaccination. Most SAEs and AEs of special interest ($n = 138$) occurred at lower than the expected population rates. Vascular ($n = 37$; 39.1/100,000 person-years) and nervous system disorders ($n = 31$; 31.7/100,000 person-years), immune system disorders ($n = 24$; 24.3/100,000 person-years), and infections and infestations ($n = 19$; 20.1/100,000 person-years) were the most common reported SAE categories. A limitation of the study was the single-arm design, with limited routinely collected morbidity comparator data in the study setting.

## Conclusions

We observed similar patterns of AEs as in phase 3 trials. AEs were mostly expected reactogenicity signs and symptoms. Furthermore, most SAEs occurred below expected rates. The single-dose Ad26.COV2.S vaccine demonstrated an acceptable safety profile, supporting the continued use of this vaccine in this setting.

## Trial registration

ClinicalTrials.gov NCT04838795; Pan African Clinical Trials Registry PACTR202102855526180.

## Author summary

### Why was this study done?

- While the safety of the Ad26.COV2.S vaccine was established in phase 3 clinical trials, continuous evaluation of safety in expanded populations is crucial.

- The Sisonke phase 3b implementation study enrolled almost half a million healthcare workers, providing an opportunity to further evaluate the safety of the single-dose Ad26.COV2.S vaccine.

### What did the researchers do and find?

- A total of 477,234 healthcare workers across all South African provinces received the Ad26.COV2.S vaccine between 17 February 2021 and 17 May 2021.

- Through self-reports and active case finding, adverse events, serious adverse events, and adverse events of special interest were identified.

- Overall occurrence of adverse events was low. The majority of adverse events reported were common reactogenicity signs and symptoms. Most serious adverse events and adverse events of special interest, including vascular events, immune system disorders, and deaths, occurred at lower than the expected population rates.

### What do these findings mean?

- The single-dose Ad26.COV2.S vaccine had an acceptable safety profile. This supports continued use of this vaccine in large rollout programmes.

### Introduction

South Africa is among the countries most affected globally by COVID-19, with over 266,000 excess natural deaths occurring between May 2020 and October 2021 (approximately 448 per 100,000 individuals) [1]. The single-dose Ad26.COV2.S vaccine showed efficacy in preventing symptomatic and severe COVID-19 disease in the ENSEMBLE study including in South Africa, where initially the beta variant and then the delta variant were the predominant circulating strains [2,3]. Here, an estimated 1,300 healthcare workers (HCWs) have died from COVID-19 as of September 2021 [4]. The Sisonke study, an open-label, single arm phase 3b implementation study of the single-dose Ad26.COV2.S vaccine, was conducted as an emergency intervention to protect HCWs in the face of an anticipated third COVID-19 wave, at a time when no vaccines were available through the national rollout. The study aimed to assess the safety and effectiveness of the Janssen Ad26.COV2.S vaccine among HCWs in South Africa.

The Ad26.COV2.S vaccine is compatible with standard vaccine storage and distribution channels and is therefore a practical vaccine for low- and middle-income countries or remote populations [5]. To date, approximately 30 million persons in the United States (US) and the European Union have received the Ad26.COV2.S vaccine [6]. Vaccine adverse event (AE) surveillance systems demonstrate that billions of people have safely received COVID-19 vaccines [7]. AEs following COVID-19 vaccination are generally mild, and include local reactions, such as injection site pain, redness, swelling, and systemic reactions, like fever, headache, fatigue, nausea, vomiting, and diarrhoea [8,9].

As reported by the US Centers for Disease Control and Prevention, severe or potentially life-threatening AEs are rare, and after 12.6 million doses of the Ad26.COV2.S vaccine, 38

cases of thrombosis with thrombocytopenia syndrome (TTS) and 98 cases of Guillain-Barré syndrome (GBS) were reported, while after 141 million second mRNA vaccine doses, 497 cases of myocarditis were reported [10]. Following the precautionary pause instituted by the Food and Drug Administration (FDA) in April 2021, the South African Health Products Regulatory Authority recommended a similar 2-week pause for the Sisonke study [11]. The study recommended with additional safeguards including screening and monitoring of participants at high risk of thrombosis and implementing measures to safely manage participants with TTS. Participant information sheets and informed consent forms were updated to include the newly identified AEs. Identification of such rare events illustrated that continued evaluation of the safety profile of vaccines post-licensure is crucial to accurately characterise safety and to identify very rare AEs that may not be reported in clinical trials.

The Sisonke study enrolled almost half a million HCWs, providing an opportunity to further evaluate the safety of the Ad26.COV2.S vaccine in an expanded population. Here we present the safety data.

## Methods

### Study participants

The Sisonke study is a multi-centre, open-label, single-arm phase 3b implementation study among HCWs (≥18 years) in South Africa, which is conducted in collaboration with the National Department of Health (ClinicalTrials.gov NCT04838795; Pan African Clinical Trials Registry PACTR202102855526180). All 1,250,000 HCWs targeted by phase 1 of the national COVID-19 Vaccine Rollout Strategy were invited for vaccination. To participate, HCWs were required to register on the national Electronic Vaccination Data System (EVDS) and provide electronic consent if they were age 18 or older. All eligible HCWs who registered on the EVDS and provided electronic consent for the study were eligible for enrolment. Known pregnant women and breastfeeding women were excluded due to a lack of sufficient safety data at that time. Details of the eligibility criteria are provided in the study protocol (S1 Appendix). A total of 477,234 HCWs received the Ad26.COV2.S vaccine between 17 February 2021 and 17 May 2021.

The institutional health research ethics committees of participating clinical research sites approved the study, which was overseen by the South African Health Products Regulatory Authority (Ref: 20200465). This study is reported as per the Consolidated Standards of Reporting Trials (CONSORT) guideline (S6 Appendix).

### Vaccination procedures

Participants received appointments for vaccination through the EVDS or were invited via employer lists. Vaccinations were conducted in collaboration with the National Department of Health public or private vaccination centres across all 9 South African provinces and overseen by Good Clinical Practice–trained personnel linked to one of the ENSEMBLE trial research sites. Participants received a single intramuscular injection of Ad26.COV2.S at a dose of $5 \times 10^{10}$ virus particles and were observed for AEs for 15 minutes post-vaccination, or for 30 minutes if they had a previous history of allergic reactions to vaccinations.

### AE reporting

AEs were reported into the study database via multiple streams using a hybrid surveillance system that combined passive with active reporting [12]. First, we designed an electronic case report form (CRF) (S2 Appendix). After vaccination, every participant received a text message

with COVID-19 infection prevention measures that also listed common signs and symptoms of reactogenicity and provided an AE reporting web link, which allowed participants access to the form for AE reporting. Reactogenicity events are pre-specified common AEs expected soon after vaccination and include systemic events such as headache, fever, myalgia, arthralgia, malaise, nausea, and chills, and local events such as pain, erythema, and induration. Second, healthcare providers were able to complete paper-based CRFs that were available at healthcare and vaccination facilities, which were then submitted to the Sisonke study safety desk and captured in the AE database. Third, the study team set up a safety desk call centre staffed by pharmacovigilance nurses, pharmacists, and safety physicians to assess and advise on AE reports. Contact details were advertised on the vaccination cards, shared on social media, and included in the text messages. Finally, spontaneous case reports via unsolicited communication by HCWs were captured and verified by safety desk staff. Telephone follow-up with the participant and attending healthcare provider was established as part of case investigation. Case reports for safety events of concern were collated from these telephone interviews, medical records, and results from laboratory and imaging investigations.

In addition, we actively linked EVDS data via national identification numbers with national patient-level disease databases, COVID-19 case notifications, and the national population registry to identify vaccine recipients with COVID-19 infections, COVID-19-related hospitalisations, and deaths. COVID-19 is a notifiable medical condition in South Africa, and tests conducted across laboratories are reported to the National Health Laboratory Service data system, which was used to identify seropositive Sisonke study participants via active linkage. A death notification form must be submitted to the Department of Home Affairs to obtain a death certificate. Therefore, in addition to case reports and active tracing, mortality was ascertained via linkage with the national population registry. After identification of deaths, the safety staff contacted next of kin and primary healthcare providers and solicited medical records to ascertain cause of death.

## Safety monitoring

AE reports were processed daily and screened for serious AEs (SAEs); SAEs were defined by the investigators as any AE that results in death, is life-threatening, requires inpatient hospitalisation or prolongation of existing hospitalisation, results in persistent or significant disability/incapacity, or is a congenital anomaly/birth defect. AEs of special interest (AESIs) were defined per the Brighton Collaboration list [9] (hereafter SAEs and AESIs are referred to collectively as SAEs). Anaphylaxis was adjudicated using the Brighton Collaboration and National Institute of Allergy and Infectious Diseases case definition, with cases needing to meet both definitions to be considered confirmed cases [13,14]. Seven days after reporting an AE, participants received a follow-up text message with a link to the electronic CRF. Participants reporting worsening or non-resolving symptoms were followed up by safety desk staff. After the FDA lifted the cautionary 2-week pause in vaccinations, 2 additional follow-up text messages were sent to all participants 7 and 14 days after vaccination. The texts highlighted signs/symptoms associated with TTS and provided a link to the electronic CRF. Safety staff made attempts to obtain medical records and supporting information from healthcare providers for all reported SAEs.

The protocol safety review team comprising principal investigators, safety physicians, and subject matter experts (haematologists, neurologist, allergy expert, and infectious disease specialists) provided oversight by weekly safety data review. An independent safety monitoring committee provided additional safety oversight.

## Statistical analysis

A prospective analysis plan was used in designing the study (see S3 Appendix). For descriptive statistics, counts and proportions were used for categorical variables, and medians and inter-quartile ranges for quantitative variables. The baseline characteristics sex, age, previous COVID-19 status, and presence of comorbidities were compared between participants report-ing AEs and those not reporting AEs using descriptive statistics. SAEs were summarised by MedDRA system organ class and adverse event preferred term. For selected SAEs, dispropor-tionality analysis was conducted: The observed (O) number of reported cases was compared to the expected (E) number based on background incidence rates, and the O/E ratio with 95% confidence interval was calculated. The 95% confidence intervals for the O/E ratio were calcu-lated as follows: (i) the 95% confidence interval of the observed number of events was calcu-lated assuming a chi-squared distribution, and (ii) each of the lower and upper limits of the confidence interval in (i) were divided by the number of expected events to produce the 95% confidence interval for the O/E ratio. Available background incidence rates were used includ-ing a medically insured population in South Africa (see S4 Appendix) for pulmonary embo-lism and deep venous thrombosis, a Tanzanian population-based cohort study for neurological events such as stroke, the Brighton Collaboration, and European population data-bases [10,15–18]. Person-time was accrued from the date of vaccination until death or dataset closure on 15 June 2021 (28 days after the last vaccination in the study). The incidence rates per 100,000 person-years were calculated using a Poisson model with person-years as an offset. Comparison of age-standardised mortality rates in the Sisonke study was with projected back-ground population mortality rates in South Africa as per the 2018 Medical Research Council Rapid Mortality Surveillance Report [19] and pre-COVID-19 local employee group life assur-ance data for a similar age-structured working population. Deaths were excluded from the SAE analysis and examined as a separate entity. COVID-19-related deaths are published in a separate effectiveness report [20]. There were no formal statistical methods that were used to handle missing data. All statistical analyses were conducted using Stata version 14 (StataCorp, College Station, TX, US). See S5 Appendix for the analytical code.

## Results

### Participants

The Sisonke study enrolled and vaccinated 477,234 participants from all 9 South African prov-inces between 17 February 2021 and 17 May 2021 (all 1,250,000 HCWs in the country were invited to participate). The majority were women (74.9%), the median age was 42 years (IQR 33–51), and 16.3% were older than 55 years. Previous COVID-19 infection was self-reported by 14.5% of participants. The most prevalent comorbidities were hypertension (15.6%), HIV infection (8.3%), and diabetes mellitus (5.9%) (Table 1).

### AEs reported by baseline characteristics

A total of 10,279 AEs were reported, of which 138 (1.4%) were classified as SAEs. Most AE reports (81%) were electronic self-reports. Women reported more AEs than men (2.3% versus 1.6%; $p < 0.001$), AE reporting decreased with increasing age (3.2% for 18- to 30-year-olds ver-sus 1.5% for >55-year-olds; $p < 0.001$), and participants with previous COVID-19 infection reported more AEs (2.6% versus 2.1%; $p < 0.001$). Persons with HIV (1.2% versus 2.2%; $p < 0.001$) or previous tuberculosis (0.8% versus 2.2%; $p = 0.043$) reported fewer AEs than those without, while more AEs were reported by participants with chronic lung disease compared to

**Table 1. Demographic and clinical characteristics of Ad26.COV2.S vaccine recipients in the Sisonke study.**

| Characteristic | Total number* (n = 477,234) | Number* reporting AEs | Percent (95% CI)* reporting AEs | p-Value |
|---|---|---|---|---|
| **Sex** | | | | <0.001 |
| Female | 357,481 | 8,389 | 2.3 (2.30–2.40) | |
| Male | 119,753 | 1,890 | 1.6 (1.51–1.65) | |
| **Age (median, IQR)** | 42 | 38 | 30–48 | <0.001 |
| **Age category (years)**** | | | | <0.001 |
| 18–30 | 85,486 | 2,697 | 3.2 (3.04–3.27) | |
| 31–45 | 209,376 | 4,484 | 2.1 (2.08–2.20) | |
| 46–55 | 104,078 | 1,902 | 1.8 (1.75–1.91) | |
| >55 | 78,162 | 1,195 | 1.5 (1.45–1.62) | |
| **Previous COVID-19** | | | | <0.001 |
| No | 408,085 | 8,478 | 2.1 (2.03–2.12) | |
| Yes | 69,149 | 1,801 | 2.6 (2.49–2.73) | |
| **Comorbidities** | | | | |
| Hypertension | | | | <0.001 |
| No | 402,853 | 8,846 | 2.2 (2.15–2.24) | |
| Yes | 74,381 | 1,433 | 1.9 (1.83–2.03) | |
| Diabetes | | | | 0.003 |
| No | 449,171 | 9,744 | 2.2 (2.13–2.21) | |
| Yes | 28,063 | 535 | 1.9 (1.75–2.07) | |
| HIV | | | | <0.001 |
| No | 437,848 | 9,800 | 2.2 (2.19–2.28) | |
| Yes | 39,386 | 479 | 1.2 (1.11–1.33) | |
| Cancer | | | | 0.528 |
| No | 475,870 | 10,253 | 2.2 (2.11–2.20) | |
| Yes | 1,364 | 26 | 1.9 (1.30–2.78) | |
| Previous tuberculosis | | | | 0.047 |
| No | 476,756 | 10,275 | 2.2 (2.11–2.20) | |
| Yes | 478 | 4 | 0.8 (0.31–2.21) | |
| Heart disease | | | | 0.232 |
| No | 473,804 | 10,195 | 2.2 (2.11–2.19) | |
| Yes | 3,430 | 84 | 2.4 (1.98–3.02) | |
| Chronic lung disease | | | | <0.001 |
| No | 475,501 | 10,197 | 2.1 (2.10–2.19) | |
| Yes | 1,733 | 82 | 4.7 (3.83–5.84) | |

AE, adverse event; IQR, interquartile range; 95% CI, 95% confidence interval; HIV, human immunodeficiency virus.

*Except for age, for which median or IQR is given.

**Missing values: age category, n = 132.

those without (4.7% versus 2.1%; $p < 0.001$). Proportions reporting AEs among those with other comorbidities were similar (Table 1).

Most reported AEs (n = 9,021, 81%) were reactogenicity events; the most common were headaches and body aches (including arthralgia, myalgia, and fatigue) that occurred within the first 7 days of vaccination, followed by mild injection site pain and fever (Fig 1). These events occurred predominately on the day of vaccination, reducing in frequency by day 2. A total of 2,109 AEs were not classified as reactogenicity events. Self-reported severity was mild to moderate in 67% of participants (n = 7,157), i.e., the event did not result in loss of ability to perform

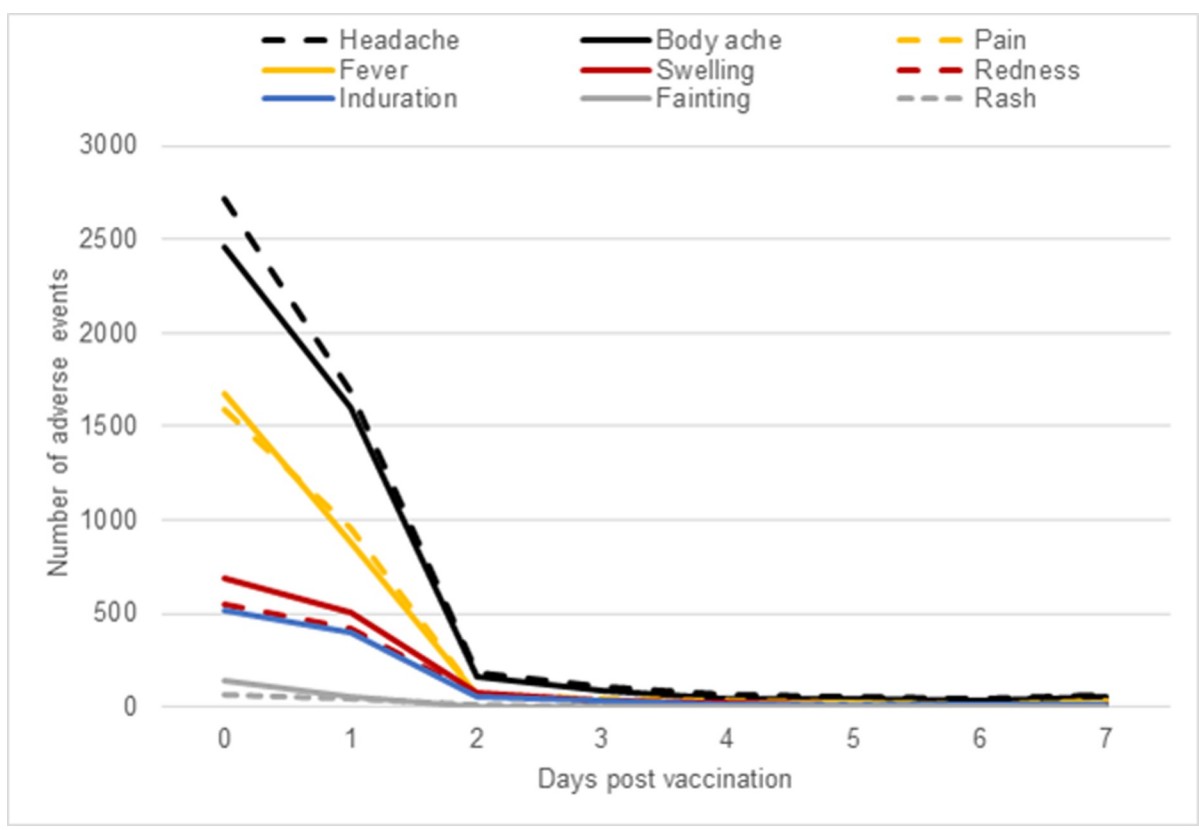

**Fig 1. Commonly occurring adverse events in the first 7 days post-vaccination.**

usual social and functional activities, while 32% of participants ($n = 3,375$) reported being unable to perform usual activities and 2% ($n = 213$) reported that they needed to visit the emergency room or were hospitalised. Follow-up at day 7 post-vaccination indicated that of the 3,831 who had responded by dataset closure, 92% (3,524) of participants reporting AEs had either completely recovered or were recovering. The remaining 8% (324) of participants were contacted by the safety team. Attempts were made by the safety team to contact all of these participants individually (3 telephone calls at least 1 day apart). Those who were contactable were interviewed via phone and, if clinically indicated, referred for further care.

## Serious adverse events

A total of 138 SAEs (excluding deaths) were reported by 136 participants (median age 42 years, IQR 35–51), with 113 (81.9%) reported by women and 25 (18.1%) by men. Most SAEs (115; 82.7%) occurred within 28 days of vaccination, with a median time to onset of 5 days (IQR 1–17) for all SAEs, and a median time of onset of 1 day (IQR 0–9) for SAEs occurring with 28 days of vaccination. Vascular ($n = 37$; 39.1/100,000 person-years) and nervous system disorders ($n = 31$; 31.7/100,000 person-years), immune system disorders ($n = 24$; 24.3/100,000 person-years), and infections and infestations ($n = 19$; 20.1/100,000 person-years) were the most common reported SAE categories (Table 2). SAE outcomes were as follows: 48 (34.8%) recovered, 36 (26.1%) recovering, and 9 (6.5%) deceased. All SAEs were followed until resolution. However, at the time of dataset closure on 15 June 2021 (28 days after the last vaccination), 45 SAEs (32.6%) were still ongoing.

**Table 2. Serious adverse events by Medical Dictionary for Regulatory Activities (MedDRA) system organ class and preferred adverse event term ($n = 138$).**

| System organ class and preferred adverse event term | N (%) | Incidence per 100,000 PY |
|---|---|---|
| **Vascular disorders** | 37 (26.8%) | 39.05 (28.30–53.90) |
| Pulmonary embolism | 10 | 10.55 (5.68–19.62) |
| Ischaemic stroke | 10 | 10.55 (5.68–19.62) |
| Deep vein thrombosis | 4 | 4.22 (1.58–11.25) |
| Acute coronary syndrome | 2 | 2.11 (0.53–8.44) |
| Hypertensive urgency | 1 | 1.06 (0.15–7.49) |
| Intracranial hypertension | 1 | 1.06 (0.15–7.49) |
| Leucocytoclastic vasculitis | 1 | 1.06 (0.15–7.49) |
| Angiosarcoma | 1 | 1.06 (0.15–7.49) |
| Retinal vein occlusion with macular haemorrhage | 1 | 1.06 (0.15–7.49). |
| Subarachnoid haemorrhage | 1 | 1.06 (0.15–7.49) |
| Cephalic vein thrombosis | 1 | 1.06 (0.15–7.49) |
| Transient thrombosis of finger | 1 | 1.06 (0.15–7.49) |
| Sagittal sinus thrombosis | 1 | 1.06 (0.15–7.49) |
| Venous sinus and cortical venous thrombosis; subarachnoid and intraparietal haemorrhage | 1 | 1.06 (0.15–7.49) |
| **Nervous system disorders** | 31 (22.5%) | 31.66 (22.14–45.29) |
| Headache | 10 | 10.55 (5.68–19.62) |
| Bell palsy | 5 | 5.28 (2.20–12.68) |
| Guillain-Barré syndrome | 4 | 4.22 (1.58–11.25) |
| Paraesthesia in lower limbs | 3 | 3.17 (1.02–9.82) |
| Ménière disease | 2 | 2.11 (0.53–8.44) |
| Seizures | 2 | 2.11 (0.53–8.44) |
| Transverse myelitis | 2 | 2.11 (0.53–8.44) |
| Chronic fatigue syndrome exacerbation | 1 | 1.06 (0.15–7.49) |
| Fibromuscular dysplasia | 1 | 1.06 (0.15–7.49) |
| Functional neurological disorder | 1 | 1.06 (0.15–7.49) |
| **Immune system disorders** | 24 (17.4%) | 24.28 (16.13–36.53) |
| Allergic reaction requiring hospitalisation | 9 | 8.44 (4.22–16.88) |
| Severe reactogenicity symptoms requiring hospitalisation | 6 | 6.33 (2.85–14.10) |
| Anaphylaxis | 4 | 4.22 (1.58–11.25) |
| Reactive arthritis | 2 | 2.11 (0.53–8.44) |
| Immune thrombocytopenic purpura | 2 | 2.11 (0.53–8.44) |
| DRESS (related to NSAID use) | 1 | 1.06 (0.15–7.49) |
| Multi-system symptoms | 1 | 1.06 (0.15–7.49) |
| **Infections and infestations** | 19 (13.8%) | 20.05 (12.79–31.44) |
| Non-COVID-19 pneumonia | 5 | 5.28 (2.20–12.68) |
| Acute appendicitis | 3 | 3.17 (1.02–9.82) |
| Meningitis | 2 | 2.11 (0.53–8.44) |
| Tuberculosis | 2 | 2.11 (0.53–8.44) |
| Respiratory tract infection | 2 | 2.11 (0.53–8.44) |
| Acute bronchitis | 1 | 1.06 (0.15–7.49) |
| Pyelonephritis | 1 | 1.06 (0.15–7.49) |

(*Continued*)

**Table 2.** (Continued)

| System organ class and preferred adverse event term | N (%) | Incidence per 100,000 PY |
|---|---|---|
| Toxoplasmosis/tuberculoma | 1 | 1.06 (0.15–7.49) |
| Interstitial pneumonitis | 1 | 1.06 (0.15–7.49) |
| Mesenteric lymphadenitis | 1 | 1.06 (0.15–7.49) |
| **Musculoskeletal and connective tissue disorders** | 5 (3.6%) | 5.28 (2.20–12.68) |
| Backache | 1 | 1.06 (0.15–7.49) |
| Knee fracture dislocation | 1 | 1.06 (0.15–7.49) |
| Disc prolapse with transient paralysis | 1 | 1.06 (0.15–7.49) |
| Rhabdomyolysis | 1 | 1.06 (0.15–7.49) |
| Transient myositis | 1 | 1.06 (0.15–7.49) |
| **Metabolism and nutritional disorders** | 4 (3.1%) | 4.22 (1.58–11.25) |
| Hypoglycaemia | 1 | 1.06 (0.15–7.49) |
| Hypoglycaemia and pneumonia | 1 | 1.06 (0.15–7.49) |
| New-onset diabetes mellitus | 1 | 1.06 (0.15–7.49) |
| Diabetic ketoacidosis | 1 | 1.06 (0.15–7.49) |
| **Gastrointestinal disorders** | 3 (2.2%) | 3.17 (1.02–9.82) |
| Acute pancreatitis | 1 | 1.06 (0.15–7.49) |
| Diarrhoea and vomiting | 1 | 1.06 (0.15–7.49) |
| Haematemesis with per rectum blood clots | 1 | 1.06 (0.15–7.49) |
| **Respiratory disorders** | 2 (1.5%) | 2.11 (0.53–8.44) |
| Acute asthma exacerbation | 1 | 1.06 (0.15–7.49) |
| Chronic bronchitis | 1 | 1.06 (0.15–7.49) |
| **Hepatobiliary disorders** | 2 (1.5%) | 2.11 (0.53–8.44) |
| Portal hypertension with upper GI bleeding | 1 | 1.06 (0.15–7.49) |
| Liver dysfunction | 1 | 1.06 (0.15–7.49) |
| **Blood and lymphatic disorders** | 2 (1.5%) | 2.11 (0.53–8.44) |
| Anaemia | 1 | 1.06 (0.15–7.49) |
| **Injury, poisoning, and procedural complications** | 1 (0.7%) | 1.06 (0.15–7.49) |
| Injection site swelling | | |
| **Psychiatric disorders** | 1 (0.7%) | 1.06 (0.15–7.49) |
| Major depressive episode | | |
| **Renal and urinary disorders** | 1 (0.7% | 1.06 (0.15–7.49) |
| Acute kidney injury | | |
| **Cardiac disorders** | 1 (0.7% | 1.06 (0.15–7.49) |
| Myocarditis recurrence | | |
| **Unclassified** | 5 (3.6%) | 5.28 (2.20–12.68) |

DRESS, drug reaction with eosinophilia and systemic symptoms; GI, gastrointestinal; NSAID, non-steroidal anti-inflammatory drug; PY, person-years.

The most common vascular disorders were pulmonary embolism (*n* = 10, 10.6 per 100,000 person years, 95% CI 5.7–19.6) and ischaemic strokes (*n* = 10, 10.6 per 100,000 person years, 95% CI 5.7–19.6), followed by deep vein thrombosis (*n* = 4, 4.2 per 100,000 person-years, 95% CI 1.6–11.3). Three participants had both pulmonary embolism and deep vein thrombosis. There were 2 cases classified as TTS. The first case was a woman in the age group 46–50 years presenting with pulmonary embolism, thrombocytopenia, and positive anti-platelet factor 4 antibody assay 9 days after vaccination. She had a history of contraceptive injection use and underlying chronic

respiratory and neurological conditions, and was being investigated for an autoimmune disorder. The second case was a woman in the age group 18–30 years who was admitted to hospital unconscious after experiencing a severe headache, restlessness, and confusion from 33 days after vaccination. A CT brain scan with venogram was in keeping with superior sagittal sinus thrombosis. Anti-platelet factor 4 antibody assay was negative, and she had marginally low platelets. She was a current smoker but had no other significant medical history. Both participants recovered.

Most events affecting the nervous system were complaints of headaches or migraines resulting in hospital admissions (*n* = 8). Five cases of Bell palsy (5.3 per 100,000 person-years, 95% CI 2.2–12.7) were reported between 1 and 42 days after vaccination; 2 men (age group 31–45 years) developed GBS about 2 weeks after vaccination, and 2 women (age group 46–55 years) developed GBS 16 and 17 days after vaccination (4.2 per 100,000 person-years, 95% CI 1.6–11.3). All participants were recovering at study end. Four cases were adjudicated as anaphylaxis by 2 physicians. All anaphylaxis cases had previous occurrence of drug- or vaccine-associated anaphylaxis and recovered fully. There was 1 case of myocarditis in a woman with previous myocarditis, which had settled prior to vaccination. She was receiving care at study end.

Table 3 summarises the disproportionality analysis that compares the occurred versus expected incidence ratio for SAEs of concern. Out of the SAEs examined, TTS seemed to occur at a rate greater than the baseline comparison, although the 95% confidence interval was wide and crossed 1 (O/E ratio 2.4, 95% CI 0.3–8.7), and GBS occurred at a rate greater than the baseline comparison population (O/E ratio 5.1, 95% CI 1.4–13.0). For the other SAEs, namely, ischaemic stroke, pulmonary embolism (non-TTS), deep vein thrombosis, acute coronary syndrome, Bell palsy, transverse myelitis, seizures, and myocarditis, the O/E ratio was less than 1. S2 Table presents the disproportionality analysis for SAEs occurring within 28 days of vaccination, and S1 Fig illustrates the frequency of SAE reporting from day of vaccination. As expected, there was a drop off in events when the analysis was restricted to 28 days post-vaccination, but there was not a significant change in the observed versus expected ratios of SAEs.

**Table 3. Observed versus expected (O/E) analysis of selected serious adverse events.**

| Adverse event | Observed count | Observed incidence rate per 100,000 PY (95% CI) | Expected count | Expected incidence rate per 100,000 PY* | O/E ratio (95% CI) |
|---|---|---|---|---|---|
| **Vascular disorders** | | | | | |
| Ischaemic stroke | 10 | 10.55 (5.68–19.62) | 102.89 | 108.60 [15] | 0.10 (0.05–0.18) |
| Pulmonary embolism | 10 | 10.55 (5.68–19.62) | 21.09 | 22.26 (S4 Appendix) | 0.47 (0.23–0.87) |
| Deep vein thrombosis | 4 | 4.22 (1.58–11.25) | 30.50 | 32.19 (S4 Appendix) | 0.13 (0.04–0.34) |
| Acute coronary syndrome | 2 | 2.11 (0.53–8.44) | 214.12 | 226.00 [16] | 0.01 (0.00–0.03) |
| Thrombosis with thrombocytopenia syndrome | 2 | 2.11 (0.53–8.44) | 0.83 | 0.88 [18] | 2.40 (0.29–8.66) |
| **Neurological disorders** | | | | | |
| Bell palsy | 5 | 5.28 (2.20–12.68) | 21.32 | 22.50 [17] | 0.23 (0.08–0.55) |
| Guillain-Barré syndrome | 4 | 4.22 (1.58–11.25) | 0.79 | 0.83 [17] | 5.09 (1.39–13.02) |
| Transverse myelitis | 2 | 2.11 (0.53–8.44) | 28.14 | 29.70 [17] | 0.08 (0.01–0.27) |
| Seizure | 2 | 2.11 (0.53–8.44) | 69.45 | 73.30 [17] | 0.03 (0.00–0.10) |
| **Cardiac disorders** | | | | | |
| Myocarditis | 1 | 1.06 (0.15–7.49) | 20.84 | 22.00 [17] | 0.05 (0.00–0.27) |

The O/E analysis compares the observed and expected numbers of cases. This may be expressed as the O/E ratio (observed incidence divided by expected incidence).

The rates are for adults (males and females combined) and are not stratified by age group. Where a range is given in the literature on incidence, the mid-point was used.

PY, person-years.

*Value followed by reference to literature from which the background incidence was derived.

A total of 157 (1.5% of participants reporting an AE) non-COVID-19-related deaths (167 per 100,000 person-years) were identified via the active linkage system with the national population registry. Of these deaths, cause of death was adjudicated for 67/157 (42.7%), and ascertainment continues for the remainder. Thirty-eight percent ($n = 60$) were male, median age was 48 years (IQR 40–57), 42 deaths (26.8%) were reported as having non-natural causes, 57% ($n = 90$) had at least 1 comorbidity; comorbidities reported were as follows: hypertension ($n = 48$, 30.6%), diabetes ($n = 32$, 20.4%), HIV ($n = 20$, 12.7%), heart disease ($n = 9$, 5.7%), and cancer ($n = 1$, 1.7%). Adjudicated causes of death included metastatic cancer ($n = 18$), HIV/AIDS-related deaths ($n = 15$), motor vehicle accidents ($n = 11$), homicide ($n = 7$), pulmonary embolism ($n = 5$), myocardial infarction ($n = 4$), cerebrovascular accident/stroke ($n = 3$), non-COVID-19 pneumonia ($n = 3$), intracerebral bleeding ($n = 3$), bleeding peptic ulcer/upper gastrointestinal tract bleeding ($n = 3$), suicide ($n = 2$), and status epilepticus ($n = 2$). Other causes are shown in S3 Table.

Nineteen deaths occurred within 28 days after vaccination. Causes were motor vehicle accident (3), upper gastrointestinal tract bleeding (3), homicide (2), HIV/AIDS-related deaths (2), and 1 each of pulmonary embolism, metastatic pancreatic cancer, drowning, dilated cardiomyopathy, renal failure, myelodysplastic syndrome, status epilepticus, suicide, and death after aortic valve and bypass surgery. A woman (age group 18–30 years) with a history of hypertension post-delivery presented 20 days after vaccination to her physician with jaundice and anuria. She then developed confusion, renal failure, and haemolysis requiring dialysis and fresh frozen plasma transfusion. She died after transfer to an intensive care unit. Investigations were in keeping thrombotic thrombocytopenic purpura (TTP). She was HIV negative, and no other triggers could be identified. Assessment of the event using the World Health Organization causality assessment tool classified this as an indeterminate temporal relationship with insufficient evidence for attribution to the vaccine [18]. In the absence of a clear alternative cause, the safety team deemed it plausible that the vaccine could have exacerbated this event in a patient with a predisposition to TTP.

Finally, we compared age-standardised mortality rates in the Sisonke study with projected background population mortality rates in South Africa as per the 2018 Medical Research Council Rapid Mortality Surveillance Report [19] and pre-COVID-19 local employee group life assurance data for a similar age-structured working population. The mortality rate in the Sisonke study was similar to the working population mortality data with similar ages, and well below that of the overall population mortality rate (Fig 2).

Age-standardised mortality rates for projected background population in South Africa as per the 2018 Medical Research Council Rapid Mortality Surveillance Report (RMS2018) [19] and pre-COVID-19 local employee group life assurance data (Group assured) for a similar age-structured working population. LL, 95% confidence interval lower limit; UL, 95% confidence interval upper limit.

## Discussion

The Sisonke study, a large single-arm, open-label phase 3b implementation study, aimed to assess the safety and effectiveness of the single-dose Ad26.COV2.S vaccine among almost half a million HCWs in South Africa. A previous study of this vaccine supported its effectiveness against severe COVID-19 disease and COVID-19-related death after vaccination, and against both the beta and delta variants [20]. With over 10,000 AE reports, to our knowledge, this was the largest safety analysis of the Ad26.COV2.S vaccine from a low- or middle-income country. As observed in phase 3 trials, similar patterns of AEs were found and were mostly expected reactogenicity signs and symptoms. Furthermore, most SAEs were rare and occurred below expected rates. However, we did observe very rare events of TTS and GBS in this study at

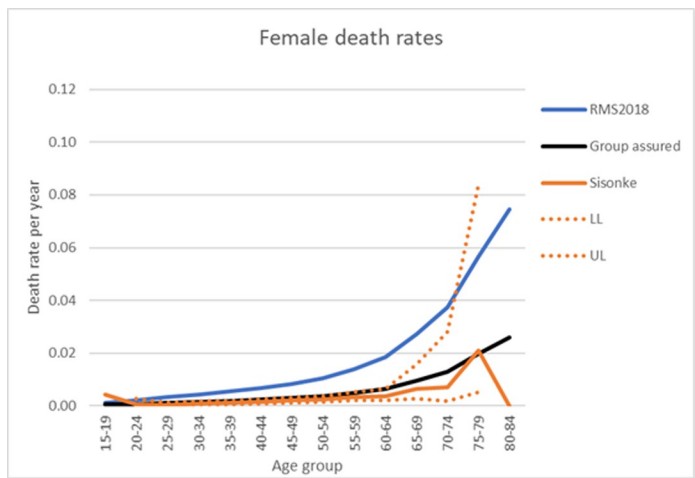
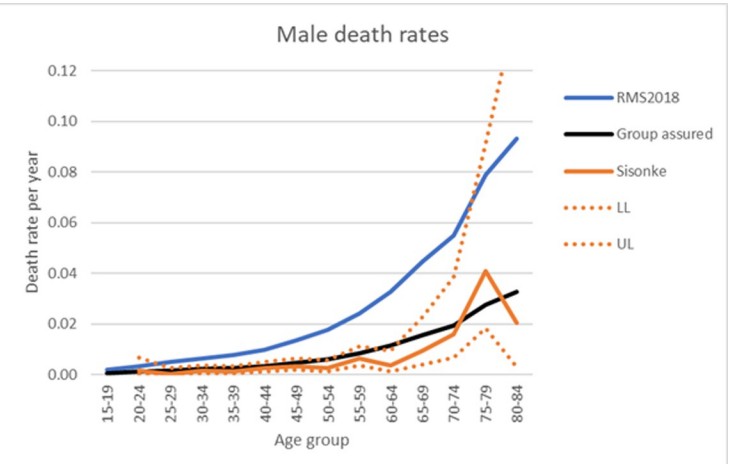

**Fig 2. Age-standardised mortality rates by sex in the Sisonke study compared to 2018 South Africa mortality rates and working population mortality rates.**

apparently higher than expected rates, though confidence intervals for these estimates were wide. Nevertheless, overall, this study provides additional real-world evidence that the vaccine is safe and well tolerated, supporting its continued use in this setting.

AEs were more often reported by women than men. While this observation may illustrate a stronger immune response in females compared to males as seen for other vaccines [21–23], behavioural factors such as reduced reporting among men may have also played a role, though these factors were not measured. The prevalence of reported AEs decreased with increasing age. A number of studies show that vaccine-related AEs and reactogenicity are less prevalent in older people due to the waning of innate immune defence mechanisms; lower systemic levels of IL-6, IL-10, and C-reactive protein; and lower neutralising antibody titres after vaccination as compared to younger individuals [24–26]. Individuals reporting previous COVID-19 infection seemed to have higher AE reporting rates. Some studies suggest that there is increased immunogenicity in the setting of past infection, leading to higher antibody titres and therefore higher reactogenicity rates [27,28].

TTS and GBS occurred at very low rates in this study; however, the disproportionality analysis showed a higher event rate than expected in the population. After 12.6 million doses of the Ad26.CoV.2 vaccine were administered in the US, 38 confirmed TTS cases and 98 GBS cases were reported [10]. Based on these data, estimates illustrate a clear advantage of vaccination despite these rare AE occurrences. For example, among women aged 30–49 years in the US, for every 6–7 cases of GBS or 8–10 cases of TTS, 10,100 COVID-19 cases, 900 hospitalisations, 140 intensive care unit admissions, and 20 deaths were prevented [10].

While the risk–benefit balance clearly favours vaccination, this study highlights the importance of ongoing safety monitoring in population-based vaccination programmes to enable continued risk–benefit assessment. The Sisonke study shows that additional surveillance, heightened awareness, and development of protocols for the management of potential clinical complications after vaccination help identify and manage possible cases early and appropriately. For example, the 2 cases of TTS were successfully managed with the support of the protocol safety review team, and both participants recovered. It is crucial that such cases are identified promptly to enable successful management. Local clinical recommendations for management of TTS were developed and implemented [29].

The Sisonke study had some limitations. First, the surveillance system was primarily passive, relying on self-reporting; thus, some AEs may have gone unreported. It is likely that the system

was better suited to detect SAEs than milder AEs, which participants may have ignored rather than reported. Second, as active contact with participants was continued up to 2 weeks post-vaccination, it is probable that SAEs other than deaths and COVID-19 events were more likely to be reported during this period, and there may have been some underestimation of SAEs that occurred later. The active linkage of the unique identifier in the EVDS with deaths in the national population registry and with COVID-19 events in the COVID-19 laboratory system ensured identification of nearly all possible deaths and COVID-19 events in the study. Third, considering the large number of participants in the study, not all self-reported AEs could be verified, and only SAEs and AEs of medical concern were investigated further. Fourth, the disproportionality analysis should be interpreted with caution given the uncertainties in both the observed and expected event rates, variable follow-up time, non–South African reference data for some groups, and potential differences in age–sex distributions between the Sisonke study and reference data. However, while disproportionality analysis in the context of safety signal detection is mainly exploratory, it has the utility of identifying potentially important associations between AEs and vaccines. In this study, the analysis confirmed current reports of the safety risk of the Ad26.COV2.S vaccine with respect to TTS and GBS [10]. Lastly, it is also important to note that without a placebo group, open-label, single-arm studies are subject to measurement bias with the potential of overreporting of AEs, and hence some caution is needed in interpreting safety data. No other safety concerns were found in this study [10,11].

Overall, the Sisonke study did not identify excess deaths in the vaccinated population compared to the general population. Mortality rates were comparable to a similar adult working population from 2018. This report illustrates the importance of accurate national mortality surveillance, especially in settings where vaccine hesitancy is driven by non-scientific and inaccurate reports in communities and through social media. The Sisonke study results are also a strong reminder that South Africa faces a large burden of disease [30]. While cancer was the most common cause of death during the study period, highlighting the urgent need for specialist oncology services, it is concerning that among HCWs advanced HIV and tuberculosis remain common causes of death. It shows that despite gains in access to HIV testing and treatment, HIV and tuberculosis care require further improvement. Local data highlight that the COVID-19 epidemic heavily impacted HIV testing and treatment initiations [31]. Motor vehicle accidents and homicides were also common causes of death, a reflection of the injury and trauma burden in South Africa. One death was related to TTP, which has previously been reported after Ad26.COV2.S vaccination and warrants further evaluation in other studies [32].

In conclusion, this study affirms that the single-dose Ad26.COV2.S vaccine is safe and well tolerated when administered to adults in South Africa. Few SAEs were observed, and they were successfully managed with prompt identification. The Sisonke study underscores the value of setting up robust pharmacovigilance systems for prompt identification, evaluation, and reporting of AEs to enable continued assessment of the risk–benefit profiles of COVID-19 vaccines. This has the potential to improve public confidence in vaccine safety and reduce vaccine hesitancy.

## Supporting information

**S1 Appendix. Sisonke study protocol.**
(PDF)

**S2 Appendix. Sisonke study adverse event case report form.**
(PDF)

**S3 Appendix. Sisonke study statistical analysis plan.**
(PDF)

**S4 Appendix. Background incidence rates for pulmonary embolism and deep vein thrombosis.**
(DOCX)

**S5 Appendix. Safety manuscript tables DO file.**
(DO)

**S6 Appendix. CONSORT checklist.**
(DOC)

**S1 Fig. Frequency of serious adverse event reports from time of vaccination.**
(TIF)

**S1 Table. Twenty most common non-reactogenicity adverse events reported.**
(DOCX)

**S2 Table. Observed versus expected analysis of selected serious adverse events within 28 days of vaccination.**
(DOCX)

**S3 Table. Mortality in the Sisonke study.**
(DOCX)

## Acknowledgments

We acknowledge the South African Medical Research Council for sponsorship and oversight, Janssen Vaccines and Prevention for the supply and transport of the study product to South Africa, Right to Care for supporting the safety monitoring and reporting infrastructure, Shirley Collie (Discovery Health) for supporting data acquisition, and the National Department of Health.

We thank all the HCWs who participated in this study, the investigators and staff members at the vaccination centres and clinical research sites, the protocol team, the protocol safety review team, and the members of the independent data and safety monitoring committee (Sipho Dlamini, Yunus Moosa, Francesca Conradie, Chris Breyer, Jeremy Nel).

**Sisonke study team:** William Brumskine, Nivashnee Naicker, Disebo Makhaza, Vimla Naicker, Logashvari Naidoo, Elizabeth Spooner, Elane van Nieuwenhuizen, Kathryn Mngadi, Maphoshane Nchabeleng, James Craig Innes, Katherine Gill, Friedrich Georg Petrick, Shaun Barnabas, Sharlaa Badal-Faesen, Sheetal Kassim, Scott Hayden Mahoney, Erica Lazarus, Anusha Nana, Rebone Molobane Maboa, Philip Kotze, Johan Lombaard, Daniel Rudolf Malan, Sheena Kotze, Phuthi Mohlala, Amy Ward, Graeme Meintjes, Dorothea Urbach, Faeezah Patel, Andreas Diacon, Khatija Ahmed, Coert Grobbelaar, Pamela Mda, Thozama Dubula, Angelique Luabeya, Musawenkosi Bhekithemba Mamba, Lesley Burgess, Rodney Dawson.

## Author Contributions

**Conceptualization:** Simbarashe Takuva, Azwidhwi Takalani, Ishen Seocharan, Nonhlanhla Yende-Zuma, Kentse Khuto, Lara Fairall, Ian Sanne, Linda Gail-Bekker, Glenda Gray, Ameena Goga, Nigel Garrett.

**Data curation:** Simbarashe Takuva, Azwidhwi Takalani, Ishen Seocharan, Nonhlanhla Yende-Zuma, Tarylee Reddy, Imke Engelbrecht, Mark Faesen, Kentse Khuto, Carmen Whyte, Veronique Bailey, Valentina Trivella, Pamela Groenewald, Ria Laubscher, Debbie Bradshaw, Harry Moultrie, Ameena Goga, Nigel Garrett.

**Formal analysis:** Simbarashe Takuva, Azwidhwi Takalani, Ishen Seocharan, Nonhlanhla Yende-Zuma, Tarylee Reddy, Imke Engelbrecht, Mark Faesen, Carmen Whyte, Veronique Bailey, Valentina Trivella, Pamela Groenewald, Rob E. Dorrington, Ria Laubscher, Debbie Bradshaw, Harry Moultrie, Ameena Goga, Nigel Garrett.

**Funding acquisition:** Lara Fairall, Ian Sanne, Linda Gail-Bekker, Glenda Gray, Ameena Goga, Nigel Garrett.

**Investigation:** Simbarashe Takuva, Azwidhwi Takalani, Imke Engelbrecht, Mark Faesen, Kentse Khuto, Carmen Whyte, Veronique Bailey, Valentina Trivella, Jonathan Peter, Jessica Opie, Vernon Louw, Pradeep Rowji, Barry Jacobson, Rob E. Dorrington, Debbie Bradshaw, Harry Moultrie, Lara Fairall, Ian Sanne, Linda Gail-Bekker, Glenda Gray, Ameena Goga, Nigel Garrett.

**Methodology:** Simbarashe Takuva, Azwidhwi Takalani, Ishen Seocharan, Nonhlanhla Yende-Zuma, Tarylee Reddy, Jonathan Peter, Jessica Opie, Vernon Louw, Pradeep Rowji, Barry Jacobson, Rob E. Dorrington, Debbie Bradshaw, Harry Moultrie, Ian Sanne, Linda Gail-Bekker, Glenda Gray, Ameena Goga, Nigel Garrett.

**Project administration:** Simbarashe Takuva, Azwidhwi Takalani, Ishen Seocharan, Imke Engelbrecht, Mark Faesen, Kentse Khuto, Carmen Whyte, Veronique Bailey, Valentina Trivella, Lara Fairall, Ian Sanne, Linda Gail-Bekker, Glenda Gray, Nigel Garrett.

**Resources:** Pamela Groenewald, Lara Fairall, Ian Sanne, Linda Gail-Bekker, Glenda Gray.

**Software:** Ishen Seocharan, Nonhlanhla Yende-Zuma, Tarylee Reddy, Ria Laubscher.

**Supervision:** Simbarashe Takuva, Azwidhwi Takalani, Jonathan Peter, Jessica Opie, Vernon Louw, Pradeep Rowji, Barry Jacobson, Linda Gail-Bekker, Ameena Goga, Nigel Garrett.

**Validation:** Simbarashe Takuva, Azwidhwi Takalani, Tarylee Reddy.

**Visualization:** Ishen Seocharan.

**Writing – original draft:** Simbarashe Takuva, Azwidhwi Takalani, Ishen Seocharan, Nonhlanhla Yende-Zuma, Tarylee Reddy, Imke Engelbrecht, Mark Faesen, Kentse Khuto, Carmen Whyte, Veronique Bailey, Valentina Trivella, Jonathan Peter, Jessica Opie, Vernon Louw, Pradeep Rowji, Barry Jacobson, Pamela Groenewald, Ria Laubscher, Debbie Bradshaw, Harry Moultrie, Ian Sanne, Linda Gail-Bekker, Glenda Gray, Ameena Goga, Nigel Garrett.

**Writing – review & editing:** Simbarashe Takuva, Azwidhwi Takalani, Ishen Seocharan, Nonhlanhla Yende-Zuma, Tarylee Reddy, Imke Engelbrecht, Mark Faesen, Kentse Khuto, Carmen Whyte, Veronique Bailey, Valentina Trivella, Jonathan Peter, Jessica Opie, Vernon Louw, Pradeep Rowji, Barry Jacobson, Pamela Groenewald, Rob E. Dorrington, Ria Laubscher, Debbie Bradshaw, Harry Moultrie, Lara Fairall, Ian Sanne, Linda Gail-Bekker, Glenda Gray, Ameena Goga, Nigel Garrett.

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
