## [Editor Report · Decision Letter 0]

1 Dec 2021

Dear Dr Takuva, 

Thank you for submitting your manuscript entitled "Safety of the single-dose Ad26.CoV2.S vaccine among healthcare workers in the phase 3b Sisonke study in South Africa" for consideration by PLOS Medicine.

Your manuscript has now been evaluated by the PLOS Medicine editorial staff and I am writing to let you know that we would like to send your submission out for external peer review.

Please re-submit your manuscript within two working days, i.e. by Dec 03 2021 11:59PM.

Kind regards,

Caitlin Moyer, Ph.D.

Associate Editor

PLOS Medicine

---

## [Decision Letter · Decision Letter 1]

23 Feb 2022

Dear Dr. Takuva,

Thank you very much for submitting your manuscript "Safety of the single-dose Ad26.CoV2.S vaccine among healthcare workers in the phase 3b Sisonke study in South Africa" (PMEDICINE-D-21-04884R1) for consideration at PLOS Medicine. 

Your paper was evaluated by a senior editor and discussed among all the editors here. It was also discussed with an academic editor with relevant expertise, and sent to three independent reviewers, including a statistical reviewer. The reviews are appended at the bottom of this email and any accompanying reviewer attachments can be seen via the link below:

[LINK]

In light of these reviews, I am afraid that we will not be able to accept the manuscript for publication in the journal in its current form, but we would like to consider a revised version that addresses the reviewers' and editors' comments. Obviously we cannot make any decision about publication until we have seen the revised manuscript and your response, and we plan to seek re-review by one or more of the reviewers. 

We expect to receive your revised manuscript by Mar 16 2022 11:59PM. Please email us (plosmedicine@plos.org) if you have any questions or concerns.

We look forward to receiving your revised manuscript. 

Sincerely,

Caitlin Moyer, Ph.D.

Associate Editor

PLOS Medicine

plosmedicine.org

1. Title: Please revise your title according to PLOS Medicine's style. Your title must be nondeclarative and not a question. It should begin with main concept if possible. "Effect of" should be used only if causality can be inferred, i.e., for an RCT. Please place the study design ("A randomized controlled trial," "A retrospective study," "A modelling study," etc.) in the subtitle (ie, after a colon).

2. Financial disclosure: Thank you for indicating that the funders had no role in the study design, data collection and analysis, decision to publish, or preparation of the manuscript. Please add to your statement in the Financial Disclosure section of the submission form: the full names of each funder, specific grant numbers, the initials of authors who received each award, and URLs to sponsors’ websites.

3. Main text: Please provide line numbers running continuously through the manuscript with the revised version.

4. Abstract: Please structure your abstract using the PLOS Medicine headings (Background, Methods and Findings, Conclusions) Please combine the Methods and Findings sections into one section, “Methods and findings”.

5. Abstract: Background: Provide a sentence or two of context of why the study is important. The final sentence should clearly state the study question.

6. Abstract: Methods and Findings: Please describe the study design, and population and setting.

7. Abstract: Methods and Findings: For the following two statements, please provide the numbers/percentages associated with these reported AEs: “The commonest reactogenicity events were headache and body aches, followed by injection site pain and fever…” and “Serious AEs and AEs of special interest including vascular and nervous system events, immune system disorders and deaths occurred at lower than the

expected population rates.”

8. Abstract: Methods and Findings: In the last sentence of the Abstract Methods and Findings section, please describe the main limitation(s) of the study's methodology.

9. Abstract: Conclusions: Please address the study implications without overreaching what can be concluded from the data; the phrase "In this study, we observed ..." may be useful.

10. Author summary: At this stage, we ask that you include a short, non-technical Author Summary of your research to make findings accessible to a wide audience that includes both scientists and non-scientists. The Author Summary should immediately follow the Abstract in your revised manuscript. This text is subject to editorial change and should be distinct from the scientific abstract. Please see our author guidelines for more information: https://journals.plos.org/plosmedicine/s/revising-your-manuscript#loc-author-summary

11. Main text: Please use square brackets for in-text citations of references, placed before the sentence punctuation, for example [1,2].

12. Background: Please title this section of the main text “Introduction” and please conclude the Introduction with a clear description of the study question or hypothesis.

13. Methods: Please report your study according to the relevant guideline, which can be found here: http://www.equator-network.org/

Please ensure that the study is reported according to the CONSORT guidelines, and please complete the CONSORT checklist and ensure that all components of CONSORT are present in the manuscript.

Please add the following statement, or similar, to the Methods: "This study is reported as per the Consolidated Standards of Reporting Trials (CONSORT) guideline (S1 Checklist)."

14. Methods: Please provide additional description of inclusion and exclusion criteria as indicated in more detail online (PACTR/clinical trials.gov registries). Please explicitly how participants were recruited, and mention that informed written consent of participants was obtained. Please comment on whether it is possible there were individuals included who had received other vaccines to prevent SARS-CoV-2 infection. Please mention/provide numbers to inform if there were eligible individuals who were contacted but declined to participate. Please include a participant flow diagram.

15. Methods: Please include the study protocol document and the analysis plan, with any amendments, as Supporting Information to be published with the manuscript if accepted.

16. Methods: Please include a copy of the electronic case report form designed for the study as a supporting information document. Please provide additional information for the verification of SAEs and AEs of medical concern, and for spontaneous reports via unsolicited HCW communication.

17. Methods: Please provide a list of those events considered as serious AEs/AEs of special interest.

18. Methods: Missing data have the potential to introduce bias in your study. Please explain how you have dealt with missing data.

19. Methods: Please describe how individual SAEs were selected for this analysis: “For selected SAEs, disproportionality analysis was conducted…”

20. Methods: Statistical analysis: “Participants reporting and not

reporting AEs were compared by baseline characteristics.” Please describe the collection of baseline characteristic data, and please describe how these data were incorporated into analyses. Please explain the categorization for age.

21. Methods: Statistical analysis: “Available background incidence rates were used including a medically insured population in South Africa (pulmonary embolism and deep venous thrombosis), Tanzanian population-based cohort study (neurological events such as stroke)

and European population databases.(10, 15–18).” Please provide more information, including how expected count/incidence were obtained for Table 3.

22. Results: Please clarify here if all post-vaccination follow up contacts provided additional data (on whether AEs resolved or not), and if not, please mention numbers lost to follow up. “Follow up at day seven post vaccination indicated that 92% of participants reporting AEs had either completely recovered or were recovering. The remaining 8% of participants were contacted by the safety team and, if required, referred for care.”

23. Results: Please present numerators and denominators for percentages, at least in the Tables if not in the main text of the Results [not necessarily each time they're mentioned].

24. Results: “One in five (19%) AEs were not consistent classified as reactogenicity events (Supplementary Table 1).” Please clarify in the text what is meant here (for example, if these were classified as serious adverse events).

25. Results/ Figure 2: Please provide additional details regarding the comparison with the local employee group life assurance data.

26. Discussion: Please be sure to present and organize the Discussion as follows: a short, clear summary of the article's findings; what the study adds to existing research and where and why the results may differ from previous research; strengths and limitations of the study; implications and next steps for research, clinical practice, and/or public policy; one-paragraph conclusion.

Comments from the reviewers:

Reviewer #1: See attachment

Michael Dewey

Reviewer #2: S. Takuva et al., reported here the results of an open label implementation study that assess the safety (and the effectiveness) of one shot of Ad26.CoV2.S vaccine of janssen among Healthcare workers (HCWs) in South Africa. This manuscript is focused on the presentattion of safety data.only. 

It is a very impresssive implementation study that included in real life conditions an half of a million of HCWs.

The manuscript is well written and easy to read. 

The introduction is concise and presented well the situation.

I have few comments:

-in the methods section, (p 5) please provide details about the exclusion of pregnant women ( there is a pregnancy test performed? what could be challenging in study of this size; if not just precise that women with a known pregnancy were excluded).

-in the results section, since the numerators are not notified in the text, please propose to the reader to see in the table 1, as it is only the percentage is available ("the majority were women (74.9%)"). In the section "SAE" of the results section page 8, it is written "114 reported by women and 25 by men" 114+25= 139 , only 138 SAE s excluding death were reported, please change. In this same paragraph, please specify at which time after vaccination the SAEs outcome were evaluated?. For the 9 death (6.5%), I suggest to precise that it represented 9/10,279 (0.09%) of people that declared an AE.

-references in the reference list are sometime not completed

This work is very interesting and indeed shows that, including in low and middle income countries, surveillance at large scale is possible, what is crucial to achieve confidence in vaccines.

Reviewer #3: I thoroughly enjoyed reading the "Safety of the single-dose Ad26.CoV2.S vaccine among healthcare workers in the phase 3b Sisonke study in South Africa" by Takuva et al. It was clearly written in excellent English, with a format and logic that was cohesive and easy to follow. Their sampling and selection criteria is clear, and their analysis follows standard methods. The reporting of their results is clear, and in their discussion and limitations they follow a logical expansion of the topic. The only suggestions that I have to improve the quality of their excellent work are as follows:

1- Given the high percentage of females in the study, a brief explanation of whether they over sampled for females would be clarifying.

2- In Tables 2 and 3, if there could be an additional column which indicates the incidence of adverse events in other large studies of the safety Ad26.CoV2.S (perhaps Ref. 7 & 8) it would allow for rapid comparison of results.

3- Several minor typo mistakes, which fall under the editorial corrections.

[LINK]

---

## [Decision Letter · Decision Letter 2]

5 May 2022

Dear Dr. Takuva,

Thank you very much for re-submitting your manuscript "Safety evaluation of the single-dose Ad26.COV2.S vaccine among healthcare workers in the Sisonke study in South Africa: a phase 3b implementation trial" (PMEDICINE-D-21-04884R2) for review by PLOS Medicine.

I have discussed the paper with my colleagues and the academic editor and it was also seen again by two reviewers. Provided that the remaining reviewer comments, and editorial and production issues are dealt with we are planning to accept the paper for publication in the journal.

[LINK]

We look forward to receiving the revised manuscript by May 12 2022 11:59PM.   

Sincerely,

Caitlin Moyer, Ph.D.

Associate Editor 

PLOS Medicine

plosmedicine.org

Requests from Editors:

1. Response to Reviewers: Please completely address the remaining point of Reviewer 1 by providing additional details on effects on participants for adverse events reported in Supporting Information Table 1.

2. Title: Please capitalize the first letter of the first word of the subtitle, and please update this in both the manuscript text and the manuscript submission system: “Safety evaluation of the single-dose Ad26.COV2.S vaccine among healthcare workers in the Sisonke study in South Africa: A phase 3b implementation trial”

3. Data availability statement: The Data Availability Statement (DAS) requires revision. If the data are owned by a third party (The National Department of Health, in this case), please note whether the data may be made available upon request or application, and state the owner of the data set and contact information for data requests (as you have done already: Office of the Director-General, E-mail: DG@health.gov.za).

We request that the analytical code be made available, without restrictions on access, in a public repository or included as Supporting Information at the time of article publication, provided it is legal and ethical to do so. We note that the Protocol and Analysis Plan are included as supporting information files already, and these files do not need to be mentioned here.

Please update this statement with information needed to access the analytical code. Please note that a study author may not serve as the point of contact for access. Please see the policy at

http://journals.plos.org/plosmedicine/s/data-availability

and FAQs at

http://journals.plos.org/plosmedicine/s/data-availability#loc-faqs-for-data-policy

4. Abstract: Methods and Findings: Please provide the trial registration information in the Abstract.

5. Abstract: Methods and Findings: Please clarify “...among all eligible HCWs in South Africa registered in the national Electronic Vaccination Data System (EVDS)…” or similar.

6. Abstract: Methods and Findings: At line 77, the number of serious AEs is reported as 139, and at line 85, this is reported as 138.

7. Abstract: Conclusions: Line 91-93: Given the finding in Table 3 for TTS and GBS, it may be more accurate to phrase this as “most SAEs occurred below expected rates.”

8. Author summary: Line 108-111: Please revise to: “The majority of adverse events reported…” and please revise the following sentence to “ Most serious adverse events…”

9. Methods: Please add the following statement, or similar, to the Methods: "This study is reported as per the Consolidated Standards of Reporting Trials (CONSORT) guideline (S1 Checklist)."

10. Methods: Line 173-174: Please reference the copy of the study protocol included as a supporting information file, and refer to it here (Supplementary Appendix 2). Please cite the online version of the protocol in the reference list, rather than including a web link here.

11. Methods: Statistical analysis: Line 239: Please mention If a prospective analysis plan was used in designing the study, for example, Supplementary Appendix 3, and please reference that file here. Please describe in the text if any changes to the analysis plan were made, and when.

12. Methods: Line 241-242: “Participants reporting and not reporting AEs were compared by baseline characteristics.” Please provide more details of this analysis.

13. Methods: Line 257-258: “COVID-related deaths were excluded in this report and published in a separate effectiveness report.” Please provide a reference to the publication if these data have been published.

14. Results: Line 266-268: We suggest rephrasing to: “...(all 1 250 000 HCWs in the country were invited to participate).” or similar.

15. Results: Line 279: Please change “less” to “fewer” AEs.

16. Results: Line 292-296: “Follow up at day seven post vaccination indicated that 92% of participants reporting AEs had either completely recovered or were recovering. The remaining 8% of participants were contacted by the safety team. Attempts were made to contact all these participants individually by the safety team (three telephonic calls at least one day apart). Those that were contactable were captured and if indicated, referred for further care.” Please include numbers in addition to reporting percentages here. Please clarify what is meant by “Those that were contactable were captured…” and please re-phrase this if possible.

17. Results: Line 296-297: “One in five (19%) AEs were not consistent classified as reactogenicity events” Please provide the actual number of AEs not classified as reactogenicity events.

18. Results: Line 307-308: “SAE outcomes were: 48 (34.8%) recovered, 36 (26.1%) recovering, 45 (32.6%) ongoing and 9 (6.5%) deceased.” Please mention the duration of follow up here, in light of the 32% of reports classified as ongoing and the following sentence mentioning events were followed up until resolution.

19. Discussion: Line 392-394: We suggest phrasing this as: “A previous study of this vaccine supported its effectiveness against severe COVID-19 disease and COVID-19-related death after vaccination, and against both beta and delta variants [20].

20. Discussion: Line 397-398: We suggest revising to: “Furthermore, most SAEs were rare and occurred below expected rates. However, we did observe very rare events of TTS and GBS in this study at apparently higher than expected rates, though confidence intervals for these estimates were wide.” or similar.

21. Discussion: Line 405: We suggest “ was lower with increasing age” instead of “reduced” here.

22. References: Please check the formatting of each reference, and please use the "Vancouver" style for reference formatting, and see our website for other reference guidelines https://journals.plos.org/plosmedicine/s/submission-guidelines#loc-references

23. When providing a DOI (e.g. for Reference 3), please give both the label and full DOI at the end of the reference. Do not provide a shortened DOI or the URL.

Please remove “[Internet]” throughout.

Please update the citation information for any preprints (e.g. reference 16).

24. Table 1: Please present p values for all comparisons. Please define all abbreviations used (AEs, IQR, HIV) in the legend.

25. Table 2: Please define abbreviations used (PY, DRESS, GI) in the legend.

26. Table 3: Please define abbreviations used (PY) in the legend. In the legend, please clarify if this should be “The rates are for adults (males and females combined) and are not stratified by age-group.”

27. Figure 1: Please provide a descriptive legend for the graph.

28. Figure 2: Please describe “UL” and “LL” as well as indicating 2018 Medical Research Council Rapid Mortality Surveillance Report (RMS2018) and pre-COVID-19 local employee group life assurance data (Group assured) in the legend.

29. CONSORT Checklist: Thank you for including the checklist. Please revise the checklist to refer to locations within the text with section and paragraph numbers (e.g. Methods, paragraph 1). Please do not refer to page numbers.

30. Supplementary Table 1: Please spell out the abbreviation “BP” in the legend.

31. Supplementary Table 2: Please define abbreviations (PY, CI, O/E) in the legend.

32. Supplementary Table 3: Please define abbreviations (IQR, HIV) in the legend.

33. Supplementary Appendix 2: Please note that pages 2-6 contain contact information (addresses/phone numbers/email addresses), please remove this information if it should not be made publicly available.

Comments from Reviewers:

Reviewer #1: The authors have addressed my points but there remains one minor issue outstanding. I asked for Supplementary Table 1 to give us more details about these other adverse events broken down by the effect on the participant (unable to perform usual activities, hospitalised). I perhaps was not clear enough as the authors' response just outlines what response the clinical and research team made to the events. This is valuable to know but not what I was asking for.

Michael Dewey

Reviewer #2: Authors correctly addressed the comments of reviewers

[LINK]

---

## [Editor Report · Decision Letter 3]

16 May 2022

Dear Dr. Takuva,

Thank you very much for re-submitting your manuscript "Safety evaluation of the single-dose Ad26.COV2.S vaccine among healthcare workers in the Sisonke study in South Africa: A phase 3b implementation trial" (PMEDICINE-D-21-04884R3) for review by PLOS Medicine.

Provided the remaining editorial and production issues are dealt with we are planning to accept the paper for publication in the journal.

[LINK]

We look forward to receiving the revised manuscript by May 18 2022 11:59PM.   

Sincerely,

Caitlin Moyer, Ph.D.

Associate Editor 

PLOS Medicine

plosmedicine.org

Requests from Editors:

1. Line 69-70: Please move the clinical trial information to the end of the abstract. “ClinicalTrials.gov number NCT04838795; Pan-African Clinical Trials registry number PACTR202102855526180.”

2. Abstract: Line 86-87: Please revise to “Most serious AEs and AEs of special interest (n=138) occurred at lower than the expected population rates.”

3. Line 90: Please spell out “SAE” at first use in the abstract.

4. Author summary: Line 116: Please re-phrase or clarify what is meant by “the large roll-out products.”

5. Line 299-300: Please remove the extraneous text from this paragraph. “One in five (19%, 2,109) AEs were not consistently classified as reactogenicity events” Please provide the actual number of AEs not classified as reactogenicity events.”

6. Table 3, Supplementary Table 2: In the table legends, please define the numbers in parentheses in the expected incidence rate columns.

7. Reference list: Please remove reference 17 or replace with another reference that is accessible. The reference list may not include unavailable and unpublished work. If relevant you may include the information or data as supplementary material or deposit the data in a publicly available database.

8. Page 17: Please also list the CONSORT Checklist, Protocol, CRF, and SAP as supporting information files.

9. Analytical code: Please rename the supporting information file with the analytical code with a more clear name and a title, and please list it at the end of the manuscript with the other supporting information files. Please refer to this file at the appropriate place in the Methods section.

[LINK]

---

## [Editor Report · Decision Letter 4]

19 May 2022

Dear Dr Takuva, 

On behalf of my colleagues and the Academic Editor, Amitabh Bipin Suthar, I am pleased to inform you that we have agreed to publish your manuscript "Safety evaluation of the single-dose Ad26.COV2.S vaccine among healthcare workers in the Sisonke study in South Africa: A phase 3b implementation trial" (PMEDICINE-D-21-04884R4) in PLOS Medicine.

Please also address the following editorial request:

Table 3 and Supporting Information Table 2: Please use square brackets to indicate references, for consistency with the main text. Please note “S4 Table” where applicable. 

PRESS

Sincerely, 

Caitlin Moyer, Ph.D. 

Associate Editor 

PLOS Medicine